# Polymethoxylated Flavones Target Cancer Stemness and Improve the Antiproliferative Effect of 5-Fluorouracil in a 3D Cell Model of Colorectal Cancer

**DOI:** 10.3390/nu11020326

**Published:** 2019-02-02

**Authors:** Carolina V. Pereira, Marlene Duarte, Patrícia Silva, Andreia Bento da Silva, Catarina M. M. Duarte, Alejandro Cifuentes, Virginia García-Cañas, Maria R. Bronze, Cristina Albuquerque, Ana Teresa Serra

**Affiliations:** 1iBET, Instituto de Biologia Experimental e Tecnológica, Apartado 12, 2780-901 Oeiras, Portugal; carolina.pereira@ibet.pt (C.V.P.); mbronze@ibet.pt (M.R.B.); 2Unidade de Investigação em Patobiologia Molecular (UIPM), Instituto Português de Oncologia de Lisboa Francisco Gentil, E.P.E (IPOLFG, EPE), 1099-023 Lisboa, Portugal; marlene_408@hotmail.com (M.D.); palsilva@hotmail.com (P.S.); 3Instituto de Tecnologia Química e Biológica António Xavier, Universidade Nova de Lisboa (ITQB NOVA), 2780-157 Oeiras, Portugal; abentosilva@ff.ulisboa.pt (A.B.d.S.); cduarte@itqb.unl.pt (C.M.M.D.); 4Faculdade de Farmácia da Universidade de Lisboa, Av das Forças Armadas, 1649-019 Lisboa, Portugal; 5Laboratory of Foodomics, Institute of Food Science Research (CIAL, CSIC), Calle Nicolás Cabrera 9, 28049 Madrid, Spain; a.cifuentes@csic.es; 6Molecular Nutrition and Metabolism, Institute of Food Science Research (CIAL, CSIC) Calle Nicolás Cabrera 9, 28049 Madrid, Spain; virginia.garcia@csic.es

**Keywords:** orange peel extract, polymethoxylated flavones, tangeretin, scutellarein tetramethylether, synergistic interactions, 5-fluorouracil, colorectal cancer, cancer stem cells, 3D cell model

## Abstract

Polymethoxylated flavones (PMFs) from citrus fruits are reported to present anticancer potential. However, there is a lack of information regarding their effect on cancer stem cell (CSC) populations, which has been recognized as responsible for tumor initiation, relapse, and chemoresistance. In this study, we evaluated the effect of an orange peel extract (OPE) and its main PMFs, namely, nobiletin, sinensetin, tangeretin, and scutellarein tetramethylether in targeting cell proliferation and stemness using a 3D cell model of colorectal cancer composed of HT29 cell spheroids cultured for 7 days in stirred conditions. Soft agar assay, ALDH1 activity, and relative quantitative gene expression analysis of specific biomarkers were carried out to characterize the stemness, self-renewal, and mesenchymal features of HT29 cell spheroids. Then, the impact of OPE and PMFs in reducing cell proliferation and modulating cancer stemness and self-renewal was assessed. Results showed that, when compared with monolayer cultures, HT29 cell spheroids presented higher ALDH1 activity (81.97% ± 5.27% compared to 63.55% ± 17.49% for 2D), upregulation of *CD44*, *PROM1*, *SOX9*, and *SNAI1* genes (1.83 ± 0.34, 2.54 ± 0.51, 2.03 ± 0.15, and 6.12 ± 1.59 times) and high self-renewal capability (352 ± 55 colonies compared to 253 ± 42 for 2D). Incubation with OPE (1 mg/mL) significantly inhibited cell proliferation and modulated cancer stemness and self-renewal ability: colony formation, ALDH1 activity, and the expression of cancer stemness biomarkers *PROM1* and *LGR5* were significantly reduced (0.66 ± 0.15 and 0.51 ± 0.14 times, respectively). Among all PMFs, tangeretin was the most efficient in targeting the CSC population by decreasing colony formation and the expression of *PROM1* and *LGR5*. Scutellarein tetramethylether was shown to modulate markers of mesenchymal/metastatic transition (increasing *CDH1* and reducing *ZEB1* and *SNAI1*) and nobiletin was capable of downregulating *PROM1* and *SNAI1* expression. Importantly, all PMFs and OPE were shown to synergistically interact with 5-fluorouracil, improving the antiproliferative response of this drug.

## 1. Introduction

Colorectal cancer (CRC) is the fourth leading cause of cancer-related deaths worldwide and, although death rates have slightly declined in the last years [1] and there has been advances in screening and surgical treatment, metastatic CRC has no known cure, and the 5-year survival rate is only 8%. Such alarming ineffectiveness of standard anticancer therapies has been attributed to the existence of relatively rare, highly drug-resistant and quiescent or slow proliferating cells with stem-like properties—cancer stem cells (CSCs). This population has the capacity to self-renew and differentiate into the spectrum of cell types observed in tumors [2,3,4], and was identified in all major human cancers [5] including CRC [6]. During the last years, intensive research has been done to characterize CSCs and discover new strategies to efficiently target this cancer cell population. Generally, CSCs have been identified by the overexpression of specific surface cell markers, including CD44, CD133, LGR5 and ALDH1. Particularly, CD44 and CD133 are surface and transmembrane glycoproteins, respectively, that have been associated with a high proliferative index and tumor progression [7,8,9]. However, studies confirmed that CD44 is a better immature cell marker than CD133 with CD44^+^/CD133^+^ population showing higher tumorigenesis that CD44^−^/CD133^+^ [10,11]. The membrane G-protein receptor LGR5 is also recognized as a CSC surface marker since its expression is restricted to the crypt bottom in normal adult colon stem cells and has been associated to the upper zones of the colonic crypt during dysplasia, suggesting an invasive cell population [12]. ALDH is a detoxifying enzyme, present in both normal and malignant colorectal cells, which is capable of conferring resistance to oxidative stress and alkylating agents, thus contributing to CSC plasticity [13,14]. In tumors, ALDH activity is more pronounced in CD44^+^ and/or CD133^+^ cells [7,13]. 

In spite of most conventional therapies failure to eliminate CSCs, this cell subpopulation represents an ideal target for novel effective strategies against CRC. In this field, food bioactive compounds may represent an attractive alternative to the conventional chemotherapeutical drugs [15]. Curcumin, resveratrol, sulforaphane, and epigallocatechin-3-gallate (EGCG) are some of the natural compounds recently reported as capable of targeting a CSC subpopulation [16] through the regulation of (i) differentiation, (ii) cell cycle, (iii) self-renewal pathways (such as Hedgehog and Wnt/β-catenin), (iv) anti- and pro-apoptotic genes, and (v) epithelial–mesenchymal transition (EMT) factors through the upregulation of epithelial genes/proteins (E-cadherin) and downregulation of mesenchymal genes/proteins (vimentin, ZEB1, SNAI1) [16,17,18,19]. In fact, recent studies have suggested the capacity of EMT to induce stemness, and vice versa, by the regulation of signaling pathways. Several hypotheses of positive, negative, or non-correlation between EMT and stemness have been discussed, however, despite EMT allowing the acquisition of migratory and invasive capacities and stemness giving cell plasticity, both phenomena may lead to chemoresistance of the cancer cells [20,21]. 

Citrus fruits are recognized to be rich sources of phytochemicals where polymethoxylated flavones (PMFs) are particular compounds of interest due to their range of biological activities, including anti-inflammation, anticancer, cardiovascular, antipathogenic, and antioxidant [22,23]. PMFs are mainly located in the citrus fruit peels [22], and their anticancer activity has been reported in vivo and in vitro in several types of tumors: skin, colon, prostate, lung, liver, and breast [24,25]. Among all PMFs, tangeretin and nobiletin have been highlighted as the most promising inhibitors of cancer cell proliferation [25]. Both phytochemicals have been reported to induce cell cycle arrest in G1 phase, and not apoptosis. Specifically, tangeretin was shown to inhibit CDK2 and CDK4 with associated increased p21 and p27 levels in the colon COLO205 cell line [26]. Nobiletin was associated with interference of metastasis through the downregulation of *MMP-7* levels in colon HT29 cell line [27]. Sinensetin is another PMF identified in citrus peels that is also reported to present antiproliferative effect in several cancer cell lines [25] and anti-angiogenesis activity [28]. 

Some emerging studies have start reporting the successful effect of natural compounds, mainly curcuminoids, terpenoids, isothiocyanates, alkaloids, and isoflavones, on targeting the expression of stemness (CD44, ALDH1A1, CD133) and metastatic biomarkers (JAK/STAT, Wnt/β-catenin, and Hedgehog signaling pathways) mostly on colon and breast CSC populations using cell and animal-based models [29]. Additionally, despite the recognized role of PMFs in the modulation of several cellular processes in CRC cells, related with tumor progression, there is a lack of information about the effect of these phytochemicals in CRC stem-like cells. In our previous work, we demonstrated the capability of a PMF-enriched OPE in inhibiting cell proliferation and reducing ALDH^+^ population in a 3D cell model of HT29 colorectal cancer cells [30], suggesting that PMFs may present a role in targeting CSCs. 

The main aim of this study was to characterize specific cell processes/signaling pathways targeted by PMF-enriched OPE, and to further investigate the effect of citrus PMFs in stemness features using a 3D cell model with CSC traits. For this purpose, a PMF-enriched extract derived from orange peels and the main PMF compounds were tested alone, and in combination, in HT29 cell spheroids developed by our group [30] that were also characterized in terms of self-renewal capability, stemness, and EMT gene expression profiles.

## 2. Materials and Methods 

### 2.1. Standard PMFs

Nobiletin, sinensetin, tangeretin, and scutellarein tetramethylether were purchased from Extrasynthese (Lyon, France). Stock solutions were prepared in DMSO (Sigma-Aldrich, St. Louis, MO, USA), and stored at 4 °C. 

### 2.2. Orange Peel Extract (OPE)

Sweet Portuguese oranges (Newhall variety) were purchased from the local supermarket in December 2016. The peels were obtained after processing the fruits into juices and then the raw material was crushed in a knife mill followed by dehydration in a freeze drier. OPE was obtained using supercritical CO_2_ and ethanol as co-solvent 20% (w/w) at 25 MPa and 45 °C, after a pre-treatment with CO_2_ during 20 min at 45 °C, under atmospheric pressure, as previously described [30]. After 30 min of extraction, the collected fraction was concentrated by rotary evaporation and a stock solution of 150 mg/mL was prepared in ethanol and stored at −20 °C until further use.

The PMF content of OPE was determined by high-performance liquid chromatography with diode array detection (HPLC-UV/DAD), as previously described [30], using a Surveyor apparatus with a diode array detector (Thermo Fisher Scientific, San Jose, CA, USA). PMF content of the extracts was determined by analyzing the peak area at 320 nm—through the data acquisition system, Chromquest 4.0 (Thermo Fisher Scientific, San Jose, CA, USA)—and comparing with the calibration curve of each compound (0.1–100 mg/L). Final results were expressed as milligrams of nobiletin, tangeretin, sinensetin, or scutellarein tetramethylether per gram of dry extract.

### 2.3. Cell Lines and Culture

The CRC cell line HT29 (ATCC, Manassas, VA, USA) was maintained in RPMI-1640 medium (Gibco, Carlsbad, CA, USA) supplemented with 10% (*v*/*v*) heat-inactivated fetal bovine serum (FBS; Biowest, Riverside, MO, USA) and kept at 37 °C in a humidified atmosphere of 5% CO_2_ in air.

### 2.4. 3D Cell Culture Using A Stirred-Tank Culture System

CRC spheroids were generated using a stirred-tank bioreactor, as previously described [30,31] with slight alterations. Briefly, HT29 single cells at 2.5 × 10^5^ cell/mL were placed in a 125 mL spinner flask (Corning^®^, NY, USA) in culture medium (RPMI-1640 supplemented with 10% (*v*/*v*) FBS). The system included a magnetic stirrer, and the culture was maintained for 7 days at 37 °C with 5% CO_2_ humidified atmosphere. The culture was started with only 60% of the final medium volume and the remainder was added after 6 h. The stirring rate was initially 40 rpm, and gradually increased to 50 and 60 rpm, after 8 and 28 h, respectively. At the 4th day post-inoculation, half of the bioreactor volume was renewed on a daily basis. Experiments were performed using spheroids collected at days 7 and 8 of culture.

### 2.5. Antiproliferative Assay in HT29 Spheroids

The antiproliferative effect of OPE and PMFs was assessed in HT29 spheroids as described elsewhere [30,32,33]. Briefly, cells were seeded at a density of 5 spheroids/well in 96-well culture plates and cell viability assessed prior to treatment (time point = 0 h) was performed using PrestoBlue^®^ reagent (Carlsbad, CA, USA), according to manufacturer’s instructions. Then, spheroids were incubated with OPE, PMFs, and 5-fluorouracil (5-FU), and the controls considered with culture medium alone and the maximum % (*v*/*v*) of solvent used. After 72 h, spheroids were washed with PBS and cell viability was assessed by PrestoBlue^®^ reagent (Carlsbad, CA, USA), as described above. Fluorescence values were analyzed in a Microplate Fluorimeter FLx800 (Bioteck Instruments, Winooski, VT, USA) (excitation and emission wavelengths of 580 nm and 595 nm, respectively). Cell viability was calculated relative to the control with culture medium (without treatment) by the following equation:
(1)$$\mathrm{Cell}\;\mathrm{viability}\;(\%)=\;\frac{\frac{\mathrm{FI}\;\mathrm{ratio}\;(\mathrm{of}\;\mathrm{treated}\;\mathrm{cells}){}_{72h}}{\mathrm{FI}\;\mathrm{ratio}\;(\mathrm{of}\;\mathrm{treated}\;\mathrm{cells}){}_{0h}}}{\frac{\mathrm{FI}\;\mathrm{ratio}\;(\mathrm{of}\;\mathrm{averaged}\;\mathrm{control}\;\mathrm{cells}){}_{72h}}{\mathrm{FI}\;\mathrm{ratio}\;(\mathrm{of}\;\mathrm{averaged}\;\mathrm{control}\;\mathrm{cells}){}_{0h}}}\times 100$$
where FI is the fluorescence intensity of HT29 spheroids at 0 h and 72 h treatment treated with OPE, PMFs, and 5-FU. Cells incubated with culture medium only and cells incubated with the maximum % (*v*/*v*) of solvent used were considered as control. At least three independent experiments were made using six replicates. 

### 2.6. Interaction Studies—Combination Assays of OPE/PMFs with 5-FU 

Combination assays were performed by treating HT29 spheroids collected at day 7 with appropriate concentrations of OPE (0.30, 0.60, 1.20, and 2.40 mg/mL), 5-FU (0.15, 0.30, 0.60, and 1.20 mg/L) and their combination with the mass ratio of 2:1 (OPE/5-FU). Isolated PMFs in the corresponding concentrations present in OPE, namely, nobiletin (14.70, 29.39, 58.78, 117.56 μM), sinensetin (13.95, 27.90, 55.79, 111.58 μM), tangeretin (3.12, 6.23, 12.46, 24.92 μM), and scutellarein tetramethylether (9.44, 18.88, 37.76, 75.52 μM), were also evaluated alone and combined with 5-FU. Cell proliferation inhibition was determined after 72 h of incubation using PrestoBlue^®^ reagent as previously described (Section 2.5). To assess synergism, antagonism, and additive effects, the combination index (CI) method was used according to Chou-Talalay equation [34]. The calculations were performed using CompuSyn software (Version 1.0, 2004, Combo Syn Inc., Paramus, NJ, USA) which considers both the potency (EC_50_) and shape of the dose–effect curve. CI < 1 indicates synergism; CI = 1 indicates an additive effect; and CI > 1 indicates antagonism (Valdés et al., 2014) [34,35]. 

### 2.7. Detection of ALDH1 Activity

HT29 spheroids were placed at a density of 50 spheroids/well in a 6-well culture and incubated with OPE and PMFs for 24 h. Spheroids incubated in culture medium and medium with the highest % (*v*/*v*) of the solvent used were considered as controls, to ensure that this solvent content did not influence ALDH1 activity. HT29 spheroids derived from spinner culture were also analyzed to characterize this 3D cell model relative to HT29 monolayer cells. All spheroids were washed in PBS and dissociated with 0.25 trypsin-EDTA (1×) (Gibco, Carlsbad, CA, USA). ALDEFLUOR™ Assay kit (STEMCELL Technologies, Vancouver, Canada) was used following manufacturer’s instructions. A negative control using diethylaminobenzaldehyde (DEAB; a specific inhibitor of ALDH1 activity) was prepared for each sample to correct the fluorescence background. Cells were sorted using CyFlow Space flow cytometer and FlowMax Software (Partec, Görlitz, Germany), by reading 10,000 events/sample at flow rate of 200–350 events/second. Data analysis was performed using Flowing Software by establishing sorting gates relative to background fluorescence of DEAB-treated samples. ALDH1 activity was normalized relative to the control without treatment/solvent. At least three independent experiments were conducted in duplicate.

### 2.8. Soft Agar Colony-Forming Unit Assay

Cell growth and proliferation under anchorage-independent conditions using soft agar assay were evaluated as described elsewhere [32,36] with slight modifications. Briefly, 6-well plates were coated with 2 mL of a solution of 1.2% low-melting agarose (Lonza, Basileia, Swiss) and 2× RPMI medium with 20% FBS at a ratio of 1:1 and allowed to rest for 1–4 h at room temperature in a sterile laminar flow hood until complete solidification of the bottom layer. Meanwhile, HT29 spheroids were dissociated, as described in Section 2.7, and the cellular suspension was adjusted to 1 × 10^3^ cell/mL in 0.3% low-melting point agarose diluted in PBS (1:1 ratio) and transferred at 2 mL/well. OPE and PMFs were added directly in this layer. Cells without treatment and cells treated with the highest % (*v*/*v*) of solvent used were considered as controls. The plates were cultured at 37 °C in 5% CO_2_ humidified atmosphere for 14 days, supplemented with 100 μL/well of RPMI containing 10% (*v*/*v*) FBS twice a week. Colonies larger than 50 μm were counted visually. Efficiency of colony formation was calculated relative to the control cells without treatment. At least two independent experiments were performed in triplicate.

### 2.9. Expression Analysis of Genes Involved in EMT, Cancer Stemness, and Wnt/β-Catenin Signaling

#### 2.9.1. RNA Extraction and Reverse Transcription

Spheroids were seeded at a density of 50 spheroids/well in a 6-well plate and treated with OPE and PMFs. Controls with ethanol or DMSO at the same % (*v*/*v*) present in extract/PMFs were considered as controls. Samples directly obtained from the spinner vessel were also assessed. After 72 h, spheroids were centrifuged (5 min, 200 *g*) and resuspended in RTL buffer (QIAGEN, Hilden, Germany) with 1% (*v*/*v*) of β-mercaptoethanol with further mechanic dissociation. RNA extraction was performed using the RNeasy^®^ Mini Kit (QIAGEN, Hilden, Germany) according to the manufacturer’s instructions. Total RNA samples were reverse transcribed into cDNA using SuperScript II Reverse Transcriptase 10,000 U (200 U/µL) in a T3 Thermocycler (Biometra, Gena, Germany), as previously described [32]. 

#### 2.9.2. Real-Time Polymerase Chain Reaction (qPCR)

qPCR reactions were performed in 96-well plates using an ABI PRISM 7000 Sequence Detection System and SDS Software (both from Applied Biosystems, Foster City, CA, USA) for determination of the threshold cycle (Ct) value (6.25 ng/µL). All reactions were carried out in triplicates towards a final volume of 15 µL containing 2 µL of cDNA. For *GAPDH*, *CDH1* (coding for the cell adhesion protein E-cadherin), *SOX9, SNAI1*, *BIRC5* (encoding the baculoviral IAP repeat-containing protein 5, also named apoptosis inhibitor survivin), and *ZEB1* genes, reactions were performed using SYBR^®^ Green PCR Master Mix (Applied Biosystems, Foster City, CA, USA). For *PROM1* (coding for CD133), *LGR5*, *CD44*, *VIM* (coding for vimentin), *CDKN1A* (coding for p21), and *CCNA2* (coding for cyclin A2) genes, reactions were performed using KAPA SYBR Fast qPCR Master Mix (2×) (Kapa Biosystems, Basileia, Swiss). Primer sequences and concentrations used are available upon request. Cycling conditions applied were 10 min at 95 °C, and 40 cycles at 95 °C for 15 s and 60 °C for 1 min. For data analysis, the expression of each target gene was normalized to the corresponding housekeeping gene levels (*GAPDH*). The overall gene expression variation was assessed by the comparative Ct (2^−ΔΔCT^) method. At least two independent experiments were performed in triplicate.

### 2.10. Statistical Analysis

All data were expressed as mean ± SD. GraphPad Prism 6 software (San Diego, CA, USA) was used to calculate EC_50_ values (the concentration of sample necessary to decrease 50% of cell population) and to analyze significant differences between data set through one-way analysis of variance (ANOVA) following Tukey’s multiple comparison tests. A *p*-value ≤ 0.05 was considered significant. 

## 3. Results and Discussion

### 3.1. Characterization of Stemness, Self-Renewal, and Mesenchymal Features of HT29 Spheroids Cultured in Stirred Conditions

In our previous study, we demonstrated that by culturing HT29 cells in a stirring-based system, for 7 days, it was possible to obtain HT29 cell spheroids with the typical characteristics of in vivo solid tumors including a necrotic/apoptotic core, hypoxia regions, presence of cancer stem cells, and a less differentiated invasive front [30,31]. In the present work, a deep evaluation of stemness and mesenchymal features was assessed and compared with HT29 monolayer cultures (Figure 1).

Results obtained by qPCR showed relevant alterations on the expression of stemness- and mesenchymal-related genes between HT29 cell monolayers and the HT29 cell spheroids (Figure 1A). Although the expression of *LGR5* decreased in the HT29 cell spheroids relative to the corresponding cell monolayer, expression levels of the stemness markers *PROM1*, *CD44*, and *SOX9* increased. In agreement, the percentage of ALDH^+^ cells on cell spheroids was also higher than in the monolayer culture (Figure 1B) confirming our previous data related with enrichment of the ALDH^+^ population during HT29 cell culture in stirred conditions [30]. These results suggest that our 3D cell model displays a cancer stem cell (CSC)-like phenotype that might recapitulate best tumor, in vivo, as demonstrated by other authors that showed an increase in ALDH^+^, CD44^+^, and CD133^+^ cells during tumor progression [13]. The lower expression of *LGR5* could be explained by expression differences along tumorigenesis stages and by a transient downregulation of Wnt/β-catenin signaling pathway, knowing that *LGR5* is a target gene of this pathway [37]. 

Gene expression analysis of mesenchymal markers *SOX9*, *SNAI1*, and *ZEB1*, known to induce EMT and mediate cell stemness features [20], showed that HT29 cell spheroids presented an increased expression of *SOX9* and *SNAI1*. These results may indicate that our 3D cell model presents a more mesenchymal and migratory tumor cell phenotype than the monolayer culture. These results are aligned with the gene expression data of stemness markers described above, which is in accordance with previous studies showing that CD44^+^ HT29 cells presented higher expression of *SNAI1* than CD44^−^ cells [38]. In fact, increased levels of CD44 and CD133 have been observed in a more mesenchymal and migratory tumor cell phenotype, and the expression of SOX9 has been reported to be activated during EMT to induce stemness [20]. On the contrary, *ZEB1* expression was lower in our 3D cell model than in the cell monolayer (Figure 1A). Studies reported SNAI1 and ZEB1 as the main regulators of EMT, however, in colorectal cancer (CRC), upregulation of *ZEB1* may occur along EMT, downstream to *SNAI1* expression. Therefore, the lower expression of *ZEB1* observed here in the 3D cell model might indicate the initial phase of EMT phenomenon with only *SNAI1* overexpression being associated [39]. Expression of *CDH1* was also evaluated (Appendix A), although no significant differences were observed between the cell monolayer and the 3D model.

Aiming at better characterizing the metastatic and tumorigenic ability of our 3D cell model, we investigated the anchorage-independent cell growth of spheroid-derived cells in a semi-solid agar matrix, to mimic some of the crucial steps of metastasis-anoikis evasion, and the proliferation in the secondary tumor site [40]. This self-renewal ability is a feature of CSCs illustrated by the ability to originate one or two stem cells by symmetrical or asymmetrical cell division [41]. In the present study, single cells from HT29 monolayer culture and from HT29 cell spheroids were placed in an agarose-based matrix and the number of new colonies formed was observed and counted. Results showed that cells from the 3D cell model originated higher number of colonies than cells from monolayer culture (Figure 1C). 

These findings, together with the overexpression of *PROM1* and *CD44*, and with the ALDH^+^-rich population in this 3D model are in agreement with an increased capability of cells from spheroids to form colonies, as suggested by other authors [42]. Therefore, it is reasonable to suggest that the self-renewal and anchorage-independent cell growth observed here might dictate the possibility of single cells derived from HT29 spheroids resembling the circulating tumor cells derived from a primary CRC, thus leading to a more aggressive phenotype characterized by anoikis evasion, metastatic, immunoresistant, chemotolerant and, especially, CSC features [39,40,42,43,44,45].

### 3.2. Characterization of the PMF Content of the OPE Extract 

The PMF-enriched extract derived from orange peels (OPE) used in this work was obtained by supercritical fluid technology with CO_2_ (80%) and ethanol (20%), according to our previous study [30]. Phytochemical characterization of the extract indicated the presence of four PMFs, namely, nobiletin, sinensetin, scutellarein tetramethylether, and tangeretin (Figure 2), at concentrations of 19.76, 17.36, 10.80, and 3.88 mg/g extract, respectively. 

It is important to mention that the concentrations of nobiletin, sinensetin, and tangeretin were lower than those obtained in our previous study (4.7, 4, and 2.5 times, respectively) which could be explained by differences in the raw material used in both studies, since it is reported that the fruit variety and the year of harvesting the oranges affect the concentration of these compounds [46] and/or by differences in the extraction process, namely, in the extraction time, as previously described [30]. 

To characterize the effect of PMF-enriched OPE and PMFs, both in isolation and in combination, in CRC cells, their antiproliferative activity, expression of cell cycle, stemness, and EMT markers, as well as anchorage-independent cell growth, were evaluated after treatment using the 3D HT29 model. As described in our previous work, HT29 3D cell spheroids display characteristics observed in in vivo carcinomas, such as the hypoxic regions, the apoptotic/necrotic core, less differentiated cells in the surrounding area [29], and higher percentage of cancer stem cells (Figure 1), which has been associated with chemotherapeutic resistance. For these reasons, the 3D cell model was selected in our study to evaluate the effect of OPE and PMFs in cancer stem cells.

### 3.3. Effect of OPE and PMFs Modulating Cell Proliferation in a 3D CRC Cell Model 

The antiproliferative effect of OPE and PMFs was evaluated in HT29 cell spheroids by analyzing cell viability after 24 and 72 h of incubation, apoptosis induction, and cell cycle arrest. As shown in Figure 3A, OPE inhibited cell proliferation in a time- and dose-dependent manner. At 72 h of incubation, OPE presented the highest antiproliferative response with an EC_50_ of 1.18 ± 0.07 mg extract/mL. PMFs alone did not present relevant inhibitory effect on cancer cell growth (Appendix A) while the mixture of the four PMF compounds (N–S–T–Sc) presented a similar response as OPE (Figure 3A), suggesting that this combination of compounds may be responsible for the antiproliferative effect. Interestingly, OPE obtained and tested in the present study, despite the 2.5–4.7× lower concentration of PMFs, presented an EC_50_ value of 2.19 ± 0.99 mg extract/mL after 24 h of incubation, less than twice the value obtained in our previous study (EC_50_ of 1.24 ± 0.15 mg extract/mL). This difference may be due to the different proportions of PMFs between the two extracts. 

To further characterize the antiproliferative effect of OPE and PMFs, the expression of the cell cycle markers *CDKN1A* and *CCNA2*, and of the apoptotic marker *BIRC5*, was also evaluated. This effect was assessed using 1 mg/mL of OPE (approximately the EC_50_ value) and equivalent concentrations of PMFs compounds present in OPE (nobiletin—49.11 μM, sinensetin—46.62 μM, tangeretin—10.41 μM and scutellarein tetramethylether—31.55 μM). In agreement with the antiproliferative effect of OPE and the mixture of PMFs, the apoptotic marker *BIRC5*, responsible for preventing apoptosis by caspase-3 and caspase-7 inhibition [47], was only downregulated when HT29 cell spheroids were treated with OPE or with the mixture of the four PMFs (N–S–T–Sc) (Figure 3B). This is even more important considering that the lower expression of *BIRC5* has been pointed out to an increase in apoptosis by inhibition of cell growth and metastasis [48]. These results are in accordance with our previous work where lower concentration (0.35 mg/mL) of a similar OPE was shown to induce apoptosis in HT29 cell spheroids through caspase-3 activation, in contrast to the main isolated PMFs that presented lower numbers of caspase-3 active cells [29]. 

The inhibition of cell growth by OPE was also revealed by the expression of *CDKN1A* and *CCNA2* genes (Figure 3C). OPE and the mixture of the four PMFs may lead to G1/S cell cycle arrest, as this has been associated to the overexpression of *CDKN1A* [49]. Moreover, treatment with the mixture of the four PMFs reduced *CCNA2* expression (Figure 3C), which is suggestive of G2/M cell-cycle arrest. Accordingly, as we have previously shown using flow cytometry analysis, a similar OPE—although at lower concentration as above mentioned—as well as the mixture of PMFs induced the cell cycle arrest of HT29 spheroids at the G2/M phase [29]. Indeed, it is known that cyclin A2 binds and activates CDC2 or CDK2 kinases, and that its expression promotes not only G2/M but also G1/S cell cycle transition [49]. The reduced expression of *CCNA2* resulting from treatment with N–S–T–Sc agrees with the reduced expression of *BIRC5* gene in response to the same treatment, in an even more pronounced way than the response to OPE (Figure 3B). In agreement, it has been reported that survivin (encoded by *BIRC5*) is expressed in the G2/M phase of the cell cycle in a cell cycle-regulated manner [50]. Therefore, cell cycle arrest in G2/M, as suggested by reduced *CCNA2* expression resultant from treatment with the four PMFs, would be associated with less survivin expression, thus explaining the observed stronger decrease of BIRC5 with this treatment. Several studies have shown that polyphenols, such as flavonoids have the capacity to induce cell cycle arrest in several cancer cells like lung, liver, prostate, breast and endothelial cells [50,51,52,53,54]. Particularly in colon cancer cells, some flavonoids were responsible for cell cycle arrest, apoptosis, and autophagy [55]. The effect of isolated PMFs on HT29 cell spheroid proliferation was negligible (Appendix A), and this effect agrees with *BIRC5*, *CDKN1A*, and *CCNA2* expression data (Figure 3B,C).

### 3.4. Effect of OPE and PMFs in Modulating Cancer Stemness and Self-Renewal in a 3D Cell Model of CRC

Aiming at evaluating and characterizing the effect of OPE and PMFs in targeting cancer stemness and metastatic potential, we analyzed (i) the expression of the cancer stemness markers, *PROM1* and *LGR5* (Figure 4A), (ii) the expression of genes encoding proteins involved in epithelial–mesenchymal transition, *SNAI1* and *ZEB1* genes (mesenchymal markers), and *CDH1* (epithelial marker) (Figure 4B), and (iii) the percentage of ALDH^+^ cells (Figure 5). In addition, self-renewal capability was assessed by soft agar assay (Figure 6). This effect was accessed using OPE and the equivalent concentrations of PMFs compounds present in OPE. 

Results showed that treatment with OPE significantly reduced the cancer stemness markers *PROM1* and *LGR5* (Figure 4A), which agrees with our results showing that OPE decreased ALDH^+^ population in a dose-dependent manner, from 0.35 to 1.8 mg/mL, with the reduction of ALDH^+^ activity being approximately 50% at the highest concentration (Figure 5A). Although OPE induced an increase of mesenchymal markers *SNAI1* and *ZEB1*, and *CDH1* expression was also increased, this does not necessarily suggest a complete EMT process (Figure 4B). Accordingly, the effect of OPE in inhibiting self-renewal ability of HT29 cell spheroids was evaluated using OPE concentrations between 0.05 and 0.35 mg/mL, with the highest concentration showing complete inhibition of colony formation (Figure 6A). This suggests that OPE was capable of reducing colony formation in a dose-dependent manner. 

Besides OPE, tangeretin was also capable of decreasing the expression of *PROM1* and *LGR5* (Figure 4A). By contrast, the mixture of PMFs (N–S–T–Sc) resulted in the overexpression of stemness markers and this effect, considering the equivalent concentration of the four PMFs in the mixture and in the extract, might be explained by antagonistic interactions among these PMFs in the absence of other compounds that may be present in OPE and that may be able to counteract this effect. It is also important to note that none of the PMFs alone was able to increase the expression of both stemness markers as observed after treatment with the PMF mixture (Figure 4A). Altogether, these findings suggest that whereas the mixture of the four PMFs is sufficient and necessary for the antiproliferative effect of OPE; for its anti-stemness effect, other compounds present in OPE are needed to counterbalance the antagonism resulting from the interaction of these PMFs. 

Regarding the effect of OPE and PMFs in the expression of EMT markers and, eventually, in the EMT process, it is important to highlight that sinensetin and scutellarein tetramethylether were the only PMFs capable of decreasing the expression of both mesenchymal markers (*SNAI1* and *ZEB1*), and to increase the expression of the epithelial marker *CDH1* (Figure 4B). These two effects have been reported to be associated with mesenchymal–epithelial transition (MET) and to a decrease in tumor aggressiveness [20]. The expression of *CDH1* gene was also significantly increased after all treatments. However, for the tested concentration of OPE, an overexpression of mesenchymal markers (*SNAI1* and *ZEB1*) was also observed, suggesting that 1 mg/mL of this extract is not effective in targeting metastatic potential of HT29 cell spheroids. A similar result was observed for the mixture of the four PMFs (Figure 4B). It is of note that this antagonistic effect does not affect the inhibition of colony formation exhibited by OPE but may eventually attenuate the corresponding effect of the PMF mixture (Figure 6A). For the isolated PMFs, nobiletin, which is the major PMF present in OPE, was the only compound capable of downregulating both a stemness and a mesenchymal marker by downregulating *PROM1* and *SNAI1*, respectively, thus suggesting that it might partially target stemness and cancer metastasis.

To further characterize the anti-stemness effect of OPE, the ALDH^+^ activity (stemness marker) was also evaluated after treatment with increasing concentrations of OPE, between 0.35 and 1.80 mg/mL. Notably, a dose dependent effect in decreasing ALDH^+^ activity was observed, from 74.4%–65.3%–67.6 % for 0.35–0.7–1.4 mg/mL, respectively, achieving its maximum of 49.9% at the maximum concentration tested (1.80 mg/mL) (Figure 5C). For this concentration, the equivalent concentrations of each PMF and their mixture were also evaluated, and only the mixture N–S–T–Sc showed a similar effect as OPE (Figure 5A). Regarding the effect of each PMF alone, nobiletin was the only compound demonstrated to significantly decrease ALDH^+^ population, probably due to having the highest content of this PMF in the extract, thus having the PMF tested at a high concentration. To test this hypothesis, all PMFs were tested at a high and equivalent concentration (100 μM), thus demonstrating that all compounds significantly decreased ALDH^+^ population relative to the control, with no significant difference between them (*p* > 0.05) (Figure 5B,C).

Inhibitors of ALDH activity had been reported, from synthetic drugs to phytochemicals such as isoflavones and terpenoids [56]. Some chemopreventive compounds, such as curcumin, piperine, and sulforaphane, had already been reported, with the effect in the stemness marker, ALDH, associated mainly to breast cancer [57,58]. These studies had 50%–90% ALDH^+^ population suppression at very low treatment dosage (1–20 μM) [57,58]. Additionally, higher concentrations than 30 μM of PEITC (phenethyl isothiocyanate) were shown to decrease ALDH population in a parallel work using the same HT29 cell spheroids [32]. 

Regarding the effect of OPE and PMFs in anchorage-independent cell growth, i.e., in metastatic capability, there was an outstanding inhibition of colony formation exhibited by OPE, in particular, in the highest OPE concentration tested (0.35 mg/mL), where no colony formation was observed. The mixture of the four PMFs at equivalent concentrations was also able to significantly (*p* < 0.0001) reduce up to 55.80% ± 7.05% the number of colonies (Figure 6A), whereas for the isolated PMFs, no inhibition was observed (Appendix A). However, when higher concentrations of PMFs were tested, all compounds showed to inhibit self-renewal ability of HT29 spheroids by reducing the number of colonies (Figure 6B). The highest effect was obtained for tangeretin, in which a concentration of 20 μM was able to reduce 51.46% ± 7.34% of colonies relative to the control. A similar effect was observed for higher doses of nobiletin (40 μM), sinensetin (60 μM), and scutellarein tetramethylether (60 μM). The higher effect of tangeretin, almost resembling that of OPE, correlates with the decrease of both cancer stemness markers *PROM1* and *LGR5* after treatment with this PMF, as well as with OPE (Figure 4A). Other studies have also shown no inhibition of colony formation when cells were treated with polymethoxyflavones [59]. Specifically regarding PMFs, nobiletin and tangeretin showed a weak effect in inhibiting colony formation in lung cancer cell lines, although this study used monolayer cells with a different type of treatment [60].

### 3.5. Effect of OPE and PMFs Combined with Chemotherapeutical Drug 5-FU

The new concept of combining chemotherapeutic drugs with food bioactive compounds has been highlighted as a very attractive field in cancer therapy since it could significantly reduce the dose of cytotoxic drugs, diminishing the adverse effects associated with chemotherapy [61]. Within this subject, we hypothesized the possibility of combining these dietary agents, such as PMFs and PMF-enriched extracts, with the chemotherapeutical drug 5-FU as a strategy for therapy of chemoresistant CRC cells. Therefore, we evaluated the antiproliferative effect of OPE/PMFs (isolated and mixed) combined with 5-FU on HT29 spheroids. The OPE/5-FU ratio used (mass ratio of 2:1) was selected based on EC_50_ values of these samples that were previously obtained in HT29 spheroids—approximately 1.2 ± 0.1 mg/mL and 0.6 ± 0.1 mg/mL, respectively, for 72 hours of incubation. For the mixture of the four PMFs and isolated compounds, the concentrations tested were equivalent to their content in OPE. The dose–response curves of PMFs/OPE and 5-FU alone, and their mixture, are compared in Figure 7 Results showed that when combined with 5-FU, all PMFs and the extract induced a higher antiproliferative response. Interaction analysis was performed using CompuSyn software, and the combination index (CI) was calculated (Table 1). As it can be deduced from the CI values lower than 1, all the OPE/PMFs and 5-FU combinations exhibit a synergistic effect.

Interestingly, the combination of scutellarein tetramethylether with 5-FU (Figure 7) showed the lowest CI (Table 1), indicating a very strong synergy and, thus, was the most promising combination. This effect could be related to the chemical structure of this compound, including the lower number of methoxy groups [62]. Future structure–activity relationship studies should be performed to determine the impact of the number and position of methoxy groups of PMFs compounds in improving the antiproliferative effectiveness of 5-FU. Additionally, dose reduction index (DRI) analysis was carried out to evaluate the magnitude (fold) of dose reduction of 5-FU. Table 1 shows DRI values for 5-FU on HT29 spheroids, demonstrating that 5-FU concentrations can be significantly reduced, especially when combined with OPE followed by the mixture N–S–T–Sc and scutellarein tetramethylether at the lower dose. 

In the literature, several polyphenols and terpenes have been shown to potentiate the effect of chemotherapeutical drugs mainly in leukemia, colon, lung, pancreas, and breast cancer, in vivo and in vitro [61,63]. In CRC, curcumin is the most commonly polyphenol studied showing synergistic interaction with 5-FU and dasatinib by sensitizing colon cancer cells, enhancing apoptosis, and inhibiting metastasis [64,65]. Also, combinations of resveratrol with oxaliplatin, and EGCG with sulindac, showed more promising anticancer effect than the compounds separately [66,67]. The results obtained in our study showed that the combination of PMFs with 5-FU might be a promising chemotherapeutic tool, probably due to the action of PMFs on cancer stemness. Future mechanistic studies should be performed in order to validate this hypothesis.

Overall, our results show that OPE targets cancer cell proliferation, stemness, metastasis, and improves the anticancer effect of 5-FU in a 3D CRC cell model with CSC-like traits. The mixture of the main PMFs (N–S–T–Sc) could be the main contributors to the inhibition of cell proliferation, cell cycle arrest, and induction of apoptosis. Among all PMFs, tangeretin and nobiletin were responsible for targeting cancer stemness through (i) downregulation of *PROM1* and *LGR5*, (ii) inhibition of colony formation, and (iii) reduction of ALDH^+^ cells, whereas sinensetin and scutellarein may play an important role in targeting cancer metastasis through the modulation of EMT genes (increasing *CDH1* and reducing *ZEB1* and *SNAI1*). Notably, and highly relevant considering the resistance to standard chemotherapeutic drugs, namely, 5-FU, exhibited by HT-29 and by a number of tumors represented by this cell line, all PMF and OPE were shown to synergistically act with 5-FU, the most used chemotherapeutic drug in CRC. In future, the impact of these phytochemicals in both monolayer cultures and 3D cell models of several colorectal cancer cell lines representative of distinct CRC subtypes (e.g., HCT116 and SW480) should be explored, to better understand the effect of OPE and PMFs in cancer stem cells.

## 4. Conclusions

To our knowledge, this is the first study recognizing the effects of citrus compounds, namely, PMFs, in modulating CSCs and metastatic potential of colorectal cancer in a 3D cellular scenario that best resembles the tumor in vivo. Overall, PMF-enriched extract derived from orange peels was demonstrated to impair cell proliferation in HT29 cell spheroids, inducing cell cycle arrest and inducing apoptosis. Moreover, it was able to target cancer stemness and self-renewal ability by reducing (i) ALDH1 activity, (ii) the expression of stemness-related genes (*PROM1, LGR5*), and (iii) the colony formation ability. Strikingly, OPE, and also the PMF mixture, highly synergistically interact with the chemotherapeutical drug 5-FU, demonstrating that 5-FU concentrations can be significantly reduced. Among all PMFs, tangeretin was the most effective in targeting cancer stemness and self-renewal ability, whereas scutellarein tetramethylether was shown to modulate EMT-related markers (increasing *CDH1* and reducing *ZEB1* and *SNAI1* expression) and highly synergistically interact with 5-FU. 

Altogether, our research provides new insights on CRC therapy, using PMFs and PMF-enriched citrus extracts to target CSCs by impairing cell proliferation, gene expression, and self-renewal potential. Additionally, the synergistic effect observed for PMFs and 5-FU may represent an exciting starting point to open the door for new therapeutic approaches, based on the use of these natural compounds, in colorectal cancer treatment. 

## Figures and Tables

**Figure 1 nutrients-11-00326-f001:**
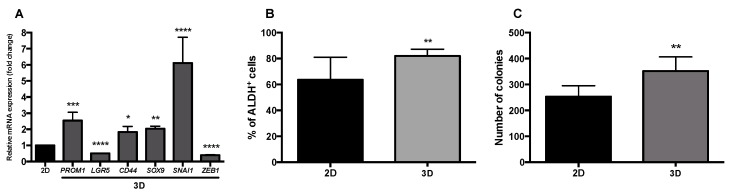
Characterization of 3D cell model (HT29 spheroids) on day 7 of culture. (**A**) Relative mRNA expression of stemness (*PROM1*, *LGR5*, *CD44*, *SOX9*) and mesenchymal (*SNAI1*, *ZEB1*) markers by qPCR. Results were normalized relative to the HT29 monolayer cells and expressed as mean ± SD of three independent experiments. (**B**) Percentage of ALDH^+^ cells. Results expressed as mean ± SD of at least three independent experiments. (**C**) Capacity of HT29 cells forming secondary colonies. After 14 days, the resulting visible colonies were counted and expressed in mean ± SD of three independent experiments. Statistically significant differences were calculated according to one-way ANOVA for multiple comparisons by Tukey’s method (* *P* ≤ 0.05, ** *P* ≤ 0.01, *** *P* ≤ 0.001, **** *P* ≤ 0.0001).

**Figure 2 nutrients-11-00326-f002:**
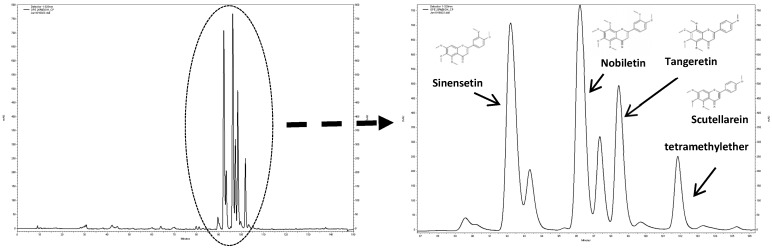
Phytochemical characterization of orange peel extract (OPE) (2 mg/mL) by HPLC-UV/DAD chromatogram recorded at 320 nm.

**Figure 3 nutrients-11-00326-f003:**
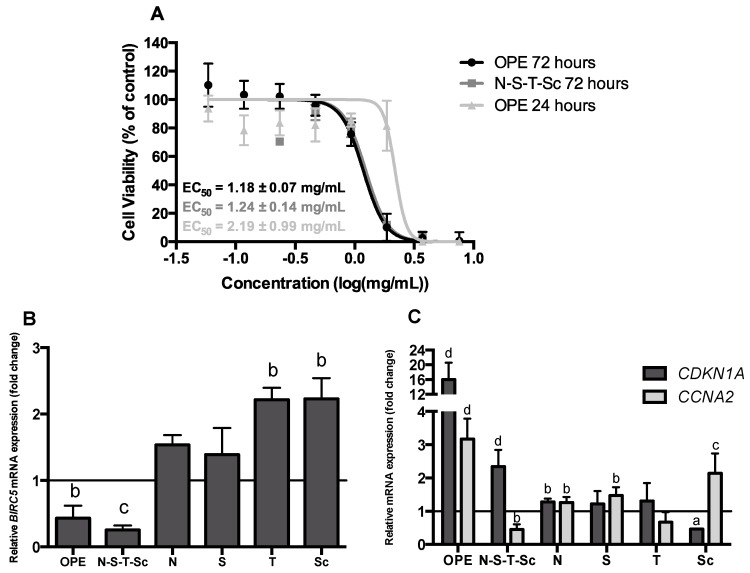
Effect of OPE and polymethoxylated flavones (PMFs) in HT29 cell spheroid proliferation. (**A**) Antiproliferative effect of OPE (24 and 72 h) and the mixture of four PMFs (tested in the same concentration as in the extract—72 h). Data are mean ± SD of four independent experiments performed with six replicates. (**B**) Relative mRNA expression of the apoptosis marker *BIRC5* in HT29 cell spheroids treated with OPE (1.00 mg/mL, ~EC_50_), the mixture of four PMFs and the isolated compounds (nobiletin—49.11 μM, sinensetin—46.62 μM, tangeretin—10.41 μM, and scutellarein tetramethylether—31.55 μM) for 72 h. Results were normalized relative to the control (untreated spheroids) and expressed in mean ± SD of two independent experiments. (**C**) Relative mRNA expression of cell cycle markers (*CDKN1A*, *CCNA2*) in HT29 cell spheroids treated with OPE (1.00 mg/mL), the mixture of four PMFs and the isolated compounds (nobiletin—49.11 μM, sinensetin—46.62 μM, tangeretin—10.41 μM, and scutellarein tetramethylether—31.55 μM) for 72 h. Results were normalized relative to the control (untreated spheroids) and expressed in mean ± SD of at least two independent experiments. Statistically significant differences were calculated according to one-way ANOVA for multiple comparisons by Tukey’s method (**a**
*P* ≤ 0.05, **b**
*P* ≤ 0.01, **c**
*P* ≤ 0.001, **d**
*P* ≤ 0.0001). Legend: N—nobiletin; S—sinensetin; T—tangeretin; Sc—scutellarein tetramethylether.

**Figure 4 nutrients-11-00326-f004:**
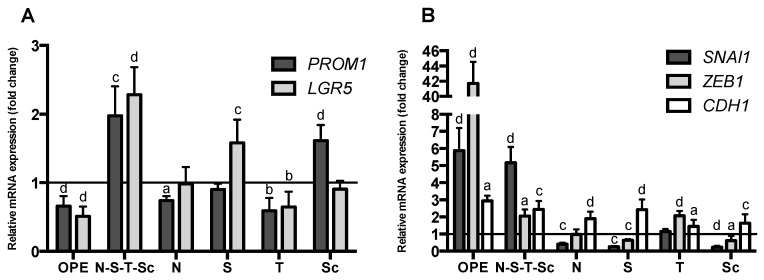
Effect of OPE (1 mg/mL) and PMFs (nobiletin—49.11 μM, sinensetin—46.62 μM, tangeretin—10.41 μM, and scutellarein tetramethylether—31.55 μM) in mRNA expression levels of stemness. (**A**) RNA expression of *PROM1* and *LGR5* genes. (**B**) RNA expression of *SNAI1*, *ZEB1*, and *CDH1* genes. Assays were performed by qPCR and results were normalized relative to the control (untreated spheroids) and expressed as mean ± SD of at least two independent experiments. Statistically significant differences were calculated according to one-way ANOVA for multiple comparisons by Tukey’s method (**a**
*P* ≤ 0.05, **b**
*P* ≤ 0.01, **c**
*P* ≤ 0.001, **d**
*P* ≤ 0.0001). Legend: N—nobiletin; S—sinensetin; T—tangeretin; Sc—scutellarein tetramethylether.

**Figure 5 nutrients-11-00326-f005:**
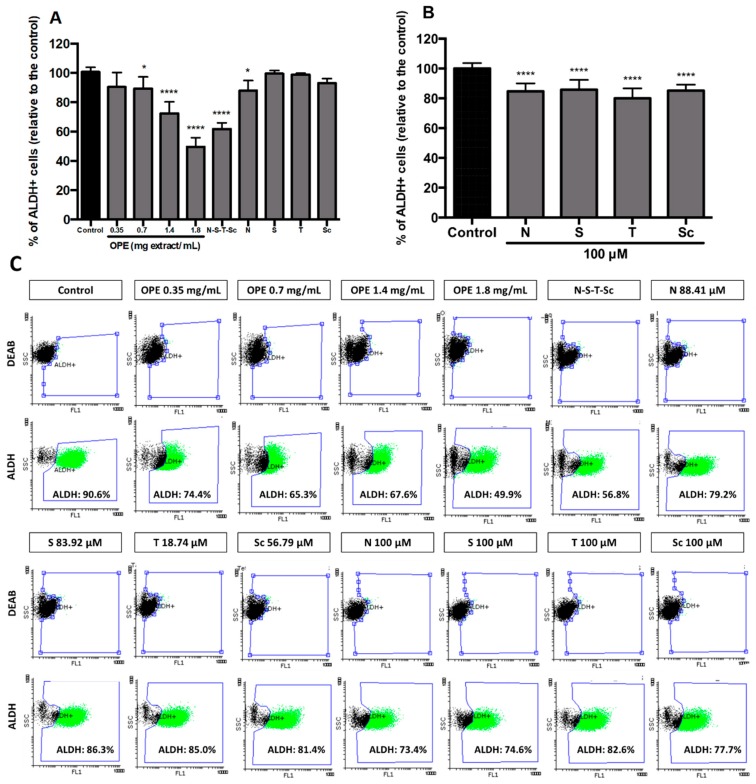
ALDH1 activity in HT29 spheroids after incubation with OPE and PMFs. (**A**) Effect of OPE (0.35, 0.7, 1.4, and 1.8 mg/mL) and PMFs in equivalent concentration present in 1.8 mg/mL of OPE (nobiletin—88.41 μM, sinensetin—83.92 μM, tangeretin—18.74 μM, and scutellarein tetramethylether—56.79 μM) in ALDH^+^ cells. (**B**) Effect of higher concentrations of PMFs (100 μM) in ALDH^+^ cells. (**C**) Representative dot-plots of ALDH1 subpopulations in HT29 spheroids from flow cytometry analysis using ALDEFLUOR^TM^ assay. All results were normalized relative to control without treatment or solvent. All data are expressed mean ± SD of at least three independent experiments. Statistically significant differences were calculated according to one-way ANOVA for multiple comparisons by Tukey’s method (* *P* ≤ 0.05, 0.001, **** *P* ≤ 0.0001). Legend: N—nobiletin; S—sinensetin; T—tangeretin; Sc—scutellarein tetramethylether.

**Figure 6 nutrients-11-00326-f006:**
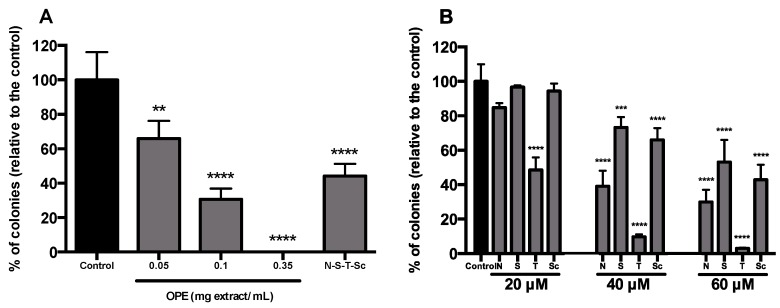
Inhibitory effect of OPE and PMFs in anchorage-independent cell growth using cells derived from HT29 spheroids. (**A**) Inhibition of colony formation by OPE (0.05, 0.1, and 0.35 mg/mL) and the mixture of PMFs in equivalent concentration present in 0.35 mg/mL of OPE (nobiletin—17.11 μM, sinensetin—16.24 μM, tangeretin—3.63 μM, and scutellarein tetramethylether—10.99 μM). (**B**) Inhibition of colony formation by PMFs (20, 40, and 60 μM). Results were normalized relative to control without treatment or solvent. All data are expressed mean ± SD of at least three independent experiments. Statistically significant differences were calculated according to one-way ANOVA for multiple comparisons by Tukey’s method (** *P* ≤ 0.01, *** *P* ≤ 0.001, **** *P* ≤ 0.0001). Legend: N—nobiletin; S—sinensetin; T— tangeretin; Sc—scutellarein tetramethylether.

**Figure 7 nutrients-11-00326-f007:**
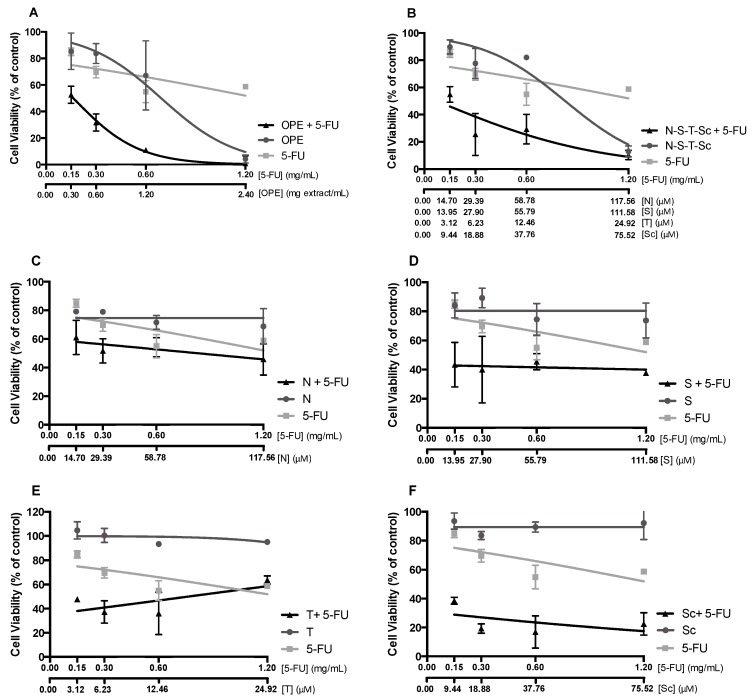
Effect of OPE or PMFs in combination with the chemotherapeutical drug 5-fluorouracil (5-FU) on HT29 spheroid proliferation after 72 h of incubation. (**A**) Antiproliferative effect of OPE and/or 5-FU. (**B**) Antiproliferative effect of N–S–T–Sc and/or 5-FU. (**C**) Antiproliferative effect of nobiletin and/or 5-FU. (**D**) Antiproliferative effect of sinensetin and/or 5-FU. (**E**) Antiproliferative effect of tangeretin and/or 5-FU. (**F**) Antiproliferative effect of scutellarein tetramethylether and/or 5-FU. The PMFs were evaluated in equivalent concentrations present in OPE concentrations tested. Data are mean ± SD of three independent experiments performed with six replicates. Legend: N—nobiletin; S—sinensetin; T—tangeretin; Sc—scutellarein tetramethylether.

**Table 1 nutrients-11-00326-t001:** Analysis of the interactions between OPE and PMFs (isolated and the mixture) with the chemotherapeutic drug 5-fluorouracil. Combination index (CI) and dose reduction index (DRI) obtained by CompuSyn software. Synergistic (CI < 0.90), additive (0.90 < CI < 1.10), and antagonistic (CI > 1.10) interactions.

Concentration	5-FU (mg/mL)	0.15	0.30	0.60	1.20
	OPE (mg/mL)	0.30	0.60	1.20	2.40
	Nobiletin (μM)	14.70	29.39	58.78	117.56
	Sinensetin (μM)	13.95	27.90	55.79	111.58
	Tangeretin (μM)	3.12	6.23	12.46	24.92
	Scutellarein tetramethylether (μM)	9.44	18.88	37.76	75.52
5-FU OPE	CI	0.31	0.52	0.52	0.46
	DRI for 5-FU	7.54	13.44	46.63	1080.81
5-FU N–S–T–Sc	CI	0.41	0.29	0.67	0.56
	DRI for 5-FU	6.61	21.06	7.97	25.60
5-FU Nobiletin	CI	0.26	0.27	0.64	0.75
	DRI for 5-FU	4.56	4.01	1.72	1.41
5-FU Sinensetin	CI	0.08	0.13	0.38	0.46
	DRI for 5-FU	12.95	7.98	2.88	2.29
5-FU Tangeretin	CI	0.17	0.22	0.43	2.72
	DRI for 5-FU	10.04	9.44	5.07	0.48
5-FU Scutellarein tetramethylether	CI	0.06	0.03	0.04	0.15
	DRI for 5-FU	17.40	35.35	22.04	6.70

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
