# Peer review of "Polymethoxylated Flavones Target Cancer Stemness and Improve the Antiproliferative Effect of 5-Fluorouracil in a 3D Cell Model of Colorectal Cancer"

_nutrients, 2019, doi:10.3390/nu11020326_

Reviewer 1 Report

In the paper the authors evaluated the effects of orange peel extract (OPE) and its main polymetoxylated flavones (PMF) tangeretin, Scutellarein tetramethylether, nobiletin  and Sinensetin

in targeting proliferation and stemnes of the HT29 colorectal cancer cell line cultured as a spheroids (3D model).    

The author demonstrated that the HT29 spheroids (respect to monolayer) presented higher ALDH activity, were enriched in cd44, prom1 sox9 and snail1 mRNA,  had an higher self renewal capability  and showed a more aggressive phenotype resembling to those of circulating tumor cells.

After this, the authors demonstrated that  OPE (1mg/ml)                                        

i)              was able to reduce  spheroid cell proliferation;

ii)             was able to modulate cancer stemness by reducing the mRNA levels of prom 1  and lgr5;

iii)            among PMF tangeretin was the most efficient in  targeting CSC population;

iv)           Scutellarein tetramethyleter modulated epithelial mesenchynal transition;

v)            All PMF and OPE  showed to synergistically act with 5-fluorouracil (5-FU).

Remarking results:

From my point of view, it is a rigorous pharmacological study that evaluates the effects of OPE in cancer growth, identifies OPE main PMF components  and  highlights the synergism between OPE and/or PMF with 5-fluorouracil.

Criticism:

If on the one hand the pharmacological study is very rigorous, the biological studies i.e.  cell cycle arrest, induction of apoptosis, modulation of stemness and mesenchymal epithelial transition are based almost exclusively on modulation of mRNA.  In my opinion the authors should reinforce their conclusion including experiments aimed to confirm  the induction of apoptosis  in terms of caspase 3 cleavage and/or annexxin V-propidium iodide staining. Furthermore the G1/S arrest could be easily demonstrated by a simple analysis of cell cycle. Considering that  some FACs analysis are already presented in the paper (i.e. ALDH activity), it should be easy for the authors perform these experiments.A second critical point is constituted by the absence of a negative control. I have no doubt that HT29 cultured in a 3D condition are enriched in "stemness"  however,  since there are no explanations about the action mechanisms of PEL and/or PMF, it could be interesting compare their action in HT29 spheroids with HT29 monolayer  for establising if  the effects are restricted (or  more pronounced)  in  cancer stem cells.   

Minor criticism:

Line 243: results and discussion (not results).

In the chapter:

3.4. Effect of OPE and PMFs in modulating cancer stemness and self-renewal in a 3D cell model of CRC

The authors discus the results reported  in figure 4 using results presented in figures 5 and 6. In order to simplify the text comprehension it shoulb be better to anticipate the results presented in these figures. 

Author Response

We thank the reviewer for his/her comments and the interest shown in this work. All comments and suggested modifications were carefully analyzed and we appreciate the opportunity to become this work clear.

Remarking results:

From my point of view, it is a rigorous pharmacological study that evaluates the effects of OPE in cancer growth, identifies OPE main PMF components and highlights the synergism between OPE and/or PMF with 5-fluorouracil.

R: We thank the reviewer for his/her comments and the interest shown in this work. All comments and suggested modifications were carefully analyzed and we appreciate the opportunity to become this work clear.

Criticism:

If on the one hand the pharmacological study is very rigorous, the biological studies i.e.  cell cycle arrest, induction of apoptosis, modulation of stemness and mesenchymal epithelial transition are based almost exclusively on modulation of mRNA.  In my opinion the authors should reinforce their conclusion including experiments aimed to confirm  the induction of apoptosis  in terms of caspase 3 cleavage and/or annexxin V-propidium iodide staining. Furthermore the G1/S arrest could be easily demonstrated by a simple analysis of cell cycle. Considering that  some FACs analysis are already presented in the paper (i.e. ALDH activity), it should be easy for the authors perform these experiments.

R: We agree with the reviewer that the biological studies of cell cycle arrest and induction of apoptosis are important to reinforce the conclusions of our study. We did not include these data as these studies were already published in our previous work (please see our paper Silva et al., 2018. Nutrition and Cancer, 70, 257-266) where it was demonstrated that an orange peel extract and the mixture of the main PMFs induced cell cycle arrest (FACs analysis) and apoptosis through caspase-3 activation. The Results and Discussion section was improved (section 3.3, lines 371-374 and 379-381) to better discuss the antiproliferative effect of PMFs and OPE.

A second critical point is constituted by the absence of a negative control. I have no doubt that HT29 cultured in a 3D condition are enriched in "stemness"  however,  since there are no explanations about the action mechanisms of PEL and/or PMF, it could be interesting compare their action in HT29 spheroids with HT29 monolayer  for establising if  the effects are restricted (or  more pronounced)  in  cancer stem cells.    

R: We thank the referee for the comment. In fact we performed additional assays in HT29 monolayer cultures showing that OPE presents higher antiproliferative effect than in HT29 cell spheroids (3D) (Figure I). This effect could be explained by the complexity of the 3D network of HT29 cell spheroids resulting on diffusion limitations of the main phytochemicals and was described in our previous work (Silva et al., 2018. Nutrition and Cancer, 70, 257-266) HT29 3D cell spheroids display characteristics observed in in vivo carcinomas, such as the hypoxic regions, the apoptotic/necrotic core, less differentiated cells in the surrounding area and higher percentage of cancer stem cells (Figure 1-uploaded), which has been associated with chemotherapeutic resistance. For these reasons, the 3D cell model was selected in our study to evaluate the effect of OPE and PMFs in cancer stem cells.

Although we consider that the Reviewer’s suggestion to include comparison studies of the action of OPE and PMFs in HT29 spheroids with HT29 monolayers is pertinent, in our opinion, these are out of the main scope of this manuscript and deserve per se to be reported in another publication addressing specifically the impact of these phytochemicals in both monolayer cultures and 3D cell models of several colorectal cancer cell lines  representative of distinct molecular subtypes (e.g. HT29, HCT116 and SW480).

Section 3 was reviewed to better clarify the rational of our work (lines 325-330, 564-568). 

Line 243: results and discussion (not results).

R: The title of section 3 was amended, accordingly.

In the chapter:

3.4. Effect of OPE and PMFs in modulating cancer stemness and self-renewal in a 3D cell model of CRC

The authors discus the results reported in figure 4 using results presented in figures 5 and 6. In order to simplify the text comprehension it shoulb be better to anticipate the results presented in these figures. 

R: We agree with the reviewer. For a better comprehension of the results, we reviewed this paragraph (Lines 399-402).

Reviewer 2 Report

The authors have done a very good study on how PMFs affect a 3D cell model of colorectal cancer. The logic is very good and experiments are done well. The results are presented well. 

Figure 5C needs to be explained in the text.

English language needs to be improved throughout the manuscript. There are many grammatical errors that need to be corrected.

A graphic model or summary should be included in the end. This will make it better for readers to understand the study.

Author Response

The authors have done a very good study on how PMFs affect a 3D cell model of colorectal cancer. The logic is very good and experiments are done well. The results are presented well. 

R: We thank the reviewer for his/her comments and the interest shown in this work. All comments and suggested modifications were carefully analyzed and we appreciate the opportunity to improve the quality of this work. 

Figure 5C needs to be explained in the text.

R: A better explanation of the data shown in Figure 5C is now included (lines 450-452 and 459).

English language needs to be improved throughout the manuscript. There are many grammatical errors that need to be corrected.

R: We thank the reviewer for the comment. The text of the manuscript was revised for typos and inaccuracies.

A graphic model or summary should be included in the end. This will make it better for readers to understand the study

R: We added a paragraph with a summary of our work (lines 554-564).

Round  2

Reviewer 1 Report

After the revision,

I have no further comments for the authors.